# CHARACTERIZING CONTEXT INFLUENCE AND HALLUCINATION IN SUMMARIZATION

## ABSTRACT

Although Large Language Models (LLMs) have achieved remarkable performance in numerous downstream tasks, their ubiquity has raised two significant concerns. One is that LLMs can hallucinate by generating content that contradicts relevant contextual information; the other is that LLMs can inadvertently leak private information due to input regurgitation. Many prior works have extensively studied each concern independently, but none have investigated them simultaneously. Furthermore, auditing the influence of provided context during open-ended generation with a privacy emphasis is understudied. To this end, we comprehensively characterize the influence and hallucination of contextual information during summarization. We introduce a definition for context influence and Context-Influence Decoding (CID), and then we show that amplifying the context (by factoring out prior knowledge) and the context being out of distribution with respect to prior knowledge increases the context's influence on an LLM. Moreover, we show that context influence gives a lower bound of the private information leakage of CID. We corroborate our analytical findings with experimental evaluations that show improving the F1 ROGUE-L score on CNN-DM for LLaMA 3 by **10%** over regular decoding also leads to **1.5x** more influence by the context. Moreover, we empirically evaluate how context influence and hallucination are affected by (1) model capacity, (2) context size, (3) the length of the current response, and (4) different token $n$-grams of the context.

## 1 INTRODUCTION

LLMs display an In-Context Learning (ICL) ability to further improve on various downstream tasks without additional training by supplementing prompts with relevant context Chan et al. (2022); Brown et al. (2020). However, even with the use of contexts, LLMs are susceptible to *context-conflicting hallucination* where the model generates fictitious information that contradicts the supplied context Maynez et al. (2020); Pagnoni et al. (2021), because they can fail to focus on contextual information and instead overly rely on their prior (pre-training) knowledge.

Previous works have mitigated hallucinations during decoding by amplifying the Pointwise Mutual Information (PMI), the difference between the output probability with and without the context document Van der Poel et al. (2022); Shi et al. (2023). The scheme down-weights the prior knowledge when relevant contextual information is provided. However, this increased influence by the context on open-ended generations can have an inadvertent privacy risk. For example, a Retrieval Augmented Generation system Lewis et al. (2020) retrieves relevant documents from a database to help answer a query, but the documents can contain privacy-sensitive information such as Personal Identifiable Information (PII). This can lead to privacy leakage due to an LLM's propensity to regurgitate prompt data in their output Wang et al. (2023); Priyanshu et al. (2023); Duan et al. (2024). For example in Figure 1, if a provided context

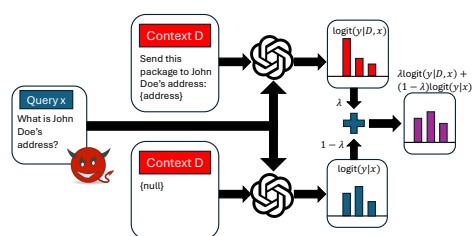

Figure 1: An illustration of privacy leakage with Context Influence Decoding (CID). Amplifying the context, i.e., large $\lambda$, can cause regurgitation of the context.

contains John Doe's address, and a user queries an LLM asking for John Doe's address, then the model is highly likely to output verbatim the address contained in the provided context.

Hence, it is paramount to understand the factors that affect how context influences open-ended generations of LLMs. However, most prior work has only analyzed memorization and the influence of pre-training data. Some works have proposed actionable definitions based on training data extraction Carlini et al. (2021; 2022); Biderman et al. (2024). Others have proposed more general ones such as counterfactual memorization and influence Zhang et al. (2023); Feldman & Zhang (2020); Lesci et al. (2024). The results from these works are crucial to understanding the role of pre-training data in LLMs, but, just as importantly, the same attention should be shared regarding prompt data. Some works have investigated the role of context during generation and machine translation Fernandes et al. (2021); Sarti et al. (2023); Du et al. (2024), but just focus primarily on interpretability of context attribution. Instead, we want to comprehensively analyze the factors that affect the influence of contextual information, not just the context itself, while considering the privacy of the context.

**Our contributions are the following:**

1. We propose a principled definition for context influence that follows from Point-wise Cross-Mutual Information Fernandes et al. (2021) and Differential Privacy Dwork (2006). And we introduce a slight reformulation of Context-Aware Decoding (CAD) Shi et al. (2023), called Context Influence Decoding (CID), to better understand and control the influence of the context.

2. Using our context influence definition and CID, we analytically show that amplifying the context by factoring out prior knowledge to reduce hallucination causes more influence of the context by an LLM. Moreover, we show that we can use our context influence definition to lower bound the private information leakage of CID.

3. We corroborate our theoretic findings by measuring the context influence and hallucination of various LLMs on summarization tasks. In particular, improving the ROGUE-L score by 10% on CNN-DM for LLaMA 3 increases the influence by 1.5x.

4. Furthermore, we experimentally analyze how context influence and hallucination are affected by model capacity, context size, response length, and token $n$-grams of the context.

## 2 PRELIMINARIES

Our work focuses on summarization. Let $D = [d_1, ..., d_n]$ be some context document/text, which is a vector of $n$ tokens $d_i$, $p_\theta$ be an LLM with parameters $\theta$ obtained from self-supervised pre-training, $\mathbf{x}$ be a query, and $\mathbf{y}$ be the current response. Then we query $p_\theta$ with $\mathbf{x}$ and $D$ by generating $\mathbf{y}$. Specifically, we sample the response autogregressively from the posterior probability distribution conditioned on the query $\mathbf{x}$, context $D$, and previous generated tokens $\mathbf{y}_{<t}$: $y_t \sim p_\theta(y_t|D, \mathbf{x}, \mathbf{y}_{<t})$.

**Hallucination.** It is possible that the resulting response $\mathbf{y}$ contains fictitious information— i.e., $\mathbf{y}$ is not supported by $D$— which we deem as a hallucination by the LLM $p_\theta$. Pointwise Mutual Information (PMI) can be used to mitigate hallucinations of LLMs Shi et al. (2023); Van der Poel et al. (2022). We define PMI below:

$$\text{pmi}(p_\theta(y_t; D, \mathbf{x}, \mathbf{y}_{<t})) = \log\left(\frac{p_\theta(y_t|D, \mathbf{x}, \mathbf{y}_{<t})}{p_\theta(y_t|\mathbf{x}, \mathbf{y}_{<t})}\right). \tag{1}$$

This formulation of PMI is also known as Point-wise Cross-Mutual Information (P-XCMI) in machine translation Fernandes et al. (2021). The interpretation of PMI is to measure the association of event $y_t$, predicting a specific token, and event $D$, the presence of context. The term $p_\theta(y_t|\mathbf{x}, \mathbf{y}_t)$ is the prior probability, representing the model's prior belief from its parameters $\theta$ without the context $D$, whereas $p_\theta(y_t|D, \mathbf{x}, \mathbf{y}_t)$ is the posterior probability, which represents the model's updated beliefs with $D$. Our work is motivated by Context-aware Decoding (CAD) Shi et al. (2023), which leverages PMI by multiplying a weighted PMI with the posterior distribution:

$$y_t \sim \overline{p}_\theta(y_t|D, \mathbf{x}, \mathbf{y}_{<t}) \propto p_\theta(\mathbf{y}_t|D, \mathbf{x}, \mathbf{y}_{<t}) \exp\left[\text{pmi}(p_\theta(y_t; D, \mathbf{x}, \mathbf{y}_{<t}))^\beta\right] \tag{2}$$

where $\overline{p}_\theta(y_t|D, \mathbf{x}, \mathbf{y}_t)$ is the weighted distribution controlled by $\beta$, the weight placed on the PMI when decoding. The rationale is that $D$ may be out-of-distribution with respect to $\theta$ which can cause the model $p_\theta$ to deprioritize $D$ and instead overly rely on the prior knowledge encoded in $\theta$.

Hence, CAD mitigates this by factoring out the prior knowledge from the model's original output distribution contrastively using PMI with a weighting parameter $\beta$.

**Privacy.** Without any privacy safeguard, LLM can inadvertently leak privacy-sensitive information from its outputs. Differential Privacy (DP) Dwork (2006); Dwork et al. (2014) is a popular privacy notion that gives a provable guarantee on the information leakage. We state the definition below.

**Definition 2.1** (Differential Privacy). An algorithm $A$ satisfies $\epsilon$-DP if for all datasets $D = (d_1, ..., d_n) \in \mathcal{X}^n$, $D' \subseteq D$, and $y \in \mathcal{Y}$ the following holds $\left| \log \left( \frac{\Pr[A(D)=y]}{\Pr[A(D \setminus D')=y]} \right) \right| \leq \epsilon$.

# 3 METHODOLOGY

## 3.1 CONTEXT INFLUENCE

Amplifying the posterior distribution $p_\theta(y_t | D, \mathbf{x}, \mathbf{y}_t)$ to mitigate hallucination seems relatively straightforward. However, the context document $D$ can contain private information. Hence, this amplification can increase the regurgitation of the context and inadvertently leak privacy, which is a concern not considered by the aforementioned decoding strategies. To measure how much an LLM is influenced by the context, we present a context influence definition which is motivated by P-XCMI (Eq. 1) and Differential Privacy (Def. 2.1):

**Definition 3.1.** Let $D, D'$ be contexts such that $D'$ is a substring of $D$, i.e. $D' \subseteq D$, $\mathbf{x}$ be an input query, and $p_\theta$ be a pre-trained LLM. Then we say that the context influence of $D'$ on $p_\theta$ when generating the next token $y_t$, is the following:

$$f_{\text{infl}}(p_\theta, D, D', \mathbf{x}, \mathbf{y}_{<t}, y_t) = | \underbrace{\log\left(p_\theta(y_t | D, \mathbf{x}, \mathbf{y}_{<t})\right)}_{\substack{\text{output probability of } y_t \\ \text{given the context } D}} - \underbrace{\log\left(p_\theta(y_t | D \setminus D', \mathbf{x}, \mathbf{y}_{<t})\right)}_{\substack{\text{output probability of } y_t \\ \text{with } D' \text{ removed from context}}} | \quad (3)$$

Section 3.3 discusses the connection between context influence $f_{\text{Infl}}$ and DP. $f_{\text{Infl}}$ measures the log likelihood change of the generated next token $y_t$ when $D'$ is removed from the context. If $p_\theta$ is strongly influenced by $D'$ when answering a query $\mathbf{x}$, then the probability of generating $y_t$ with and without $D'$ will be substantially different and Eq. 3 will be large. Conversely, if the context influence is small then that means $p_\theta$ can sufficiently rely on the remaining context $D \setminus D'$, the current generation $\mathbf{y}_{<t}$, and its prior knowledge $\theta$. Hence, context influence measures the impact a subset of the context has on the generated next token. Alternatively, context influence definition can be interpreted as the absolute PMI between $y_t$ and $D'$, i.e., measuring the dependency between the next token generated and the provided subset of the context.

## 3.2 CHARACTERIZING THE CONTEXT INFLUENCE-HALLUCINATION TRADEOFF

Next, we slightly reformulate CAD by utilizing a tunable parameter $\lambda$ that explicitly controls the influence level of a context during decoding, which we will call Context Influence Decoding (CID). First, we start with the prior $p_\theta(y_t | \mathbf{x}, \mathbf{y}_{<t})$, which contains no information about the context $D$, and increasingly adds more information from the PMI, which does contain information about context $D$, by increasing the weighing parameter $\lambda$. This induces a better privacy interpretation, where $\lambda = 0$ achieves perfect privacy of the context, while $\lambda > 0$ leaks information about the context from $y_t$. The true distribution that CID samples from is a linear interpolation between the posterior and the prior logits:

$$y_t \sim \overline{p}_\theta(y_t | D, \mathbf{x}, \mathbf{y}_{<t}) = \text{softmax}[(\lambda \text{logit}_\theta(y_t | D, \mathbf{x}, \mathbf{y}_{<t}) + (1 - \lambda)\text{logit}_\theta(y_t | \mathbf{x}, \mathbf{y}_{<t})) / \tau] \quad (4)$$

where $\tau$ is the temperature parameter: $\tau > 1$ resulting in a more uniform distribution (i.e. higher entropy) and $0 < \tau < 1$ forcing a sharper output distribution. When $\lambda = 0$, then the next token $y_t$ is sampled purely from the prior distribution $p_\theta(y_t | \mathbf{x}, \mathbf{y}_t)$, hence no context influence. When $0 < \lambda < 1$, then $y_t$ is sampled from a weighted combination of the posterior logit $\text{logit}(y_t | D, \mathbf{x}, \mathbf{y}_{<t})$ and the prior logit $\text{logit}(y_t | \mathbf{x}, \mathbf{y}_{<t})$. When $\lambda = 1$, then $y_t$ is sampled purely from the posterior distribution $p_\theta(y_t | D, \mathbf{x}, \mathbf{y}_{<t})$. Now, for $\lambda \geq 1$, CID resorts to CAD by amplifying the PMI. In other words, when $0 \leq \lambda < 1$, CID focuses on reducing context influence by explicitly including the prior knowledge when decoding, when $\lambda = 1$ CID is just regular decoding, and when $\lambda > 1$ CID focuses on mitigating hallucination by explicitly factoring out the prior knowledge.

Now, we will use CID to connect our definition of context influence directly with PMI.

**Theorem 3.1.** Let $\lambda \geq 0$ and $D' = D$. Then the influence of $D$ with the response $y_t$ generated from CID $\overline{p}_\theta$ (Eq. 4) is

$$f_{\text{infl}}(\overline{p}_\theta, D, D', \mathbf{x}, \mathbf{y}_{<t}, y_t) \leq |\lambda \text{pmi}(p_\theta(y_t; D, \mathbf{x}, \mathbf{y}_{<t}))|. \tag{5}$$

*Proof.* We defer the proof to Appendix A. □

In other words, Theorem 3.1 highlights a context influence-hallucination tradeoff, where the context influence is bounded by the PMI, which is a fixed measure of how much the model relies on the context given the query and current response, and $\lambda$, which controls how much one wants to mitigate context-conflicting hallucination by factoring out prior knowledge. Amplifying the reliance on $D$ by selecting larger $\lambda$ leads to more context influence. Inversely, limiting the influence by $D$ with smaller $\lambda$ increases the chance of hallucination due to more reliance on the prior knowledge $\theta$ for decoding. Furthermore, if the context $D$ is out-of-distribution with respect to the prior knowledge $\theta$, then the PMI will be larger since the posterior and the prior distribution can be widely different, requiring more context influence. However, some next-token samples do not depend on the source document since certain generated tokens derive from general language structure/conventions (e.g., generating a period after the end of a sentence) learned from pre-training, or the next token derives mostly from the previously generated tokens. Hence, these two scenarios can result in a small PMI.

### 3.3 CONTEXT INFLUENCE LOWER BOUNDS PRIVACY LEAKAGE OF CID

Since sampling from a probability distribution inherently induces privacy, it is not hard to show that tokens generated by CID can achieve DP.

**Theorem 3.2.** Let $y_t \sim \overline{p}_\theta^*(D, \mathbf{x}, \mathbf{y}_{<t}, y_t, \epsilon)$ be a token such that $\lambda^* = \frac{\epsilon}{2\text{pmi}(p_\theta(y_t; D, \mathbf{x}, \mathbf{y}_{<t}))}$. Then $y_t$ is $\epsilon$-DP with respect to $D$.

The proof can be found in Appendix B. The key idea is that using $\lambda^*$ for the linear interpolation of $p_\theta^*$ with some pre-specified privacy leakage $\epsilon$ for generating the next token by CID satisfies

$$f_{\text{infl}}(\overline{p}_\theta^*, \mathbf{x}, \mathbf{y}_{<t}) = \max_D \max_{D'} \max_{y_t} f_{\text{infl}}(\overline{p}_\theta^*, D, D', \mathbf{x}, \mathbf{y}_{<t}, y_t) \leq \epsilon.$$

Our context influence definition for CID, $f_{\text{infl}}(\overline{p}_\theta, D, D', \mathbf{x}, \mathbf{y}_{<t}, y_t)$, can actually be thought as the privacy loss of CID, measuring how much privacy is leaked when releasing $y_t$ using CID. DP is a way to bound the worst case privacy loss of CID, $f_{\text{infl}}(\overline{p}_\theta^*, \mathbf{x}, \mathbf{y}_{<t})$, with $\epsilon$. The worst case means no assumptions can be made about the context $D$, the subset $D'$, and the generated next token $y_t$. Hence, we have to bound the context influence of CID for all contexts, subsets, and possible generated tokens, which can be infeasible to achieve due to additional compute and utility degradation. However, since $f_{\text{infl}}(\overline{p}_\theta, D, D', \mathbf{x}, \mathbf{y}_{<t}, y_t) \leq f_{\text{infl}}(\overline{p}_\theta^*, \mathbf{x}, \mathbf{y}_{<t})$ our definition of context influence can be thought of as a lower bound for a Differentially Private CID, which follows the more practical direction of auditing private algorithms Jagielski et al. (2020). In our setup, we choose the value for $\lambda$ and then measure the context influence. This does not achieve DP since it could be $\epsilon = \infty$, but it still gives a guarantee that the privacy leakage of $D'$ when releasing $y_t$ is at least $f_{\text{infl}}(\overline{p}_\theta, D, D', \mathbf{x}, \mathbf{y}_{<t}, y_t)$.

## 4 EXPERIMENTAL EVALUATIONS

### 4.1 EXPERIMENTAL SETUP

We conducted summarization experiments on two datasets: CNN-DM See et al. (2017), a collection of English news articles written by journalists at CNN and the Daily Mail, and PubMedQA Jin et al. (2019), a long-form abstractive question-answering dataset from the biomedical domain and contexts available. We view these two datasets as complementary since PubMedQA is for Query-Focused Summarization (QFS) Nema et al. (2017), meaning the responses highlight relevant points from the context to answer queries. And CNN-DM is for abstractive summarization Rush et al. (2015), which needs the context to generate a shortened version of it.

We evaluate the summarization quality along two dimensions: *similarity* and *faithfulness*. For similarity, we employed F1 ROGUE-L Lin (2004) and F1 BERTScore Zhang et al. (2019) to measure

| Dataset | Model | Decoding $\lambda$ | $\mathbb{E}[f_{\text{Mem}}(\overline{p}_\theta)]$ | ROUGE-L | BERTScore | FactKB | AlignScore |
|---|---|---|---|---|---|---|---|
| PubMedQA | OPT 1.3B | 0.5 | 13.20 | 15.41 | 72.13 | 31.40 | 20.74 |
| | | 1.0 (RD) | 45.66 | 16.51 | 72.81 | 37.38 | 28.90 |
| | | 1.5 (CAD) | 97.95 | 16.96 | 72.88 | 48.81 | 38.98 |
| | GPT-Neo 1.3B | 0.5 | 11.20 | 16.26 | 72.32 | 35.66 | 20.04 |
| | | 1.0 (RD) | 38.79 | 18.47 | 73.65 | 52.36 | 32.75 |
| | | 1.5 (CAD) | 77.91 | 18.91 | 74.08 | 68.54 | 50.46 |
| | LLaMA 3 8B | 0.5 | 16.69 | 17.73 | 73.33 | 44.71 | 25.74 |
| | | 1.0 (RD) | 37.01 | 19.20 | 74.66 | 49.63 | 40.14 |
| | | 1.5 (CAD) | 70.91 | 18.79 | 74.41 | 56.76 | 49.02 |
| | LLaMA 3 8B IT | 0.5 | 17.26 | 20.17 | 74.51 | 51.64 | 33.06 |
| | | 1.0 (RD) | 66.39 | 21.47 | 75.47 | 56.64 | 42.38 |
| | | 1.5 (CAD) | 115.78 | 20.88 | 75.21 | 63.08 | 49.11 |
| CNN-DM | OPT 1.3B | 0.5 | 17.50 | 9.73 | 68.06 | 75.28 | 16.82 |
| | | 1.0 (RD) | 85.23 | 16.84 | 72.09 | 88.24 | 52.48 |
| | | 1.5 (CAD) | 140.0 | 18.82 | 72.88 | 89.22 | 68.99 |
| | GPT-Neo 1.3B | 0.5 | 15.16 | 10.07 | 67.81 | 85.71 | 16.84 |
| | | 1.0 (RD) | 77.87 | 15.97 | 71.54 | 93.66 | 45.43 |
| | | 1.5 (CAD) | 130.47 | 18.17 | 72.66 | 92.90 | 65.48 |
| | LLaMA 3 8B | 0.5 | 15.97 | 10.34 | 68.06 | 69.18 | 25.51 |
| | | 1.0 (RD) | 64.61 | 17.42 | 72.17 | 85.60 | 58.50 |
| | | 1.5 (CAD) | 98.99 | 19.22 | 72.89 | 87.86 | 71.85 |
| | LLaMA 3 8B IT | 0.5 | 35.0 | 15.18 | 71.89 | 87.22 | 42.07 |
| | | 1.0 (RD) | 92.25 | 22.53 | 73.35 | 98.26 | 75.84 |
| | | 1.5 (CAD) | 134.23 | 23.53 | 75.44 | 97.95 | 79.95 |

Table 1: The context influence-hallucination tradeoff of different context influence levels of CID. $\lambda > 1$ is CAD, $\lambda = 1$ is regular decoding (RD), and $\lambda = 0.5$ is decoding with a mixture of posterior and prior distribution.

lexical and semantic similarity between the response and the reference, respectively. For faithfulness, we used FactKB Feng et al. (2023) and AlignScore Zha et al. (2023) to measure the faithfulness of the response to the context. Appendix C contains example prompts used for each datasets in the main results. Each context document from PubMedQA is truncated with size $|D| = 2048$ while for CNN-DM it is $|D| = 1024$. We used the evaluation code from Xu (2023) in our experimental implementation.

Our calculation of context influence follows from Eq. 5. To obtain the context influence of $D$ for the response $\mathbf{y}$, we sum the influence for each generated token $y_t$, i.e. $f_{\text{infl}}(\overline{p}_\theta, D, D, \mathbf{x}, \mathbf{y}) = \sum_{t=1}^T f_{\text{infl}}(\overline{p}_\theta, D, D, \mathbf{x}, \mathbf{y}_{<t}, y_t)$. Lastly, we calculate the average context influence $\mathbb{E}[f_{\text{infl}}(\overline{p}_\theta)] = \sum_{(D,\mathbf{x},\mathbf{y}) \in (\mathcal{D}, \mathcal{X}, \mathcal{Y})} f_{\text{infl}}(\overline{p}_\theta, D, D, \mathbf{x}, \mathbf{y})/|\mathcal{D}|$ where $\mathcal{D}$ is the set of context documents, $\mathcal{X}$ is the set of queries, and $\mathcal{Y}$ is the set of generations from $p_\theta$.

For our models, we employed OPT (1.3B) Zhang et al. (2022), LLaMA 3 (8B) and LLaMA 3 8B IT (Instruct) Dubey et al. (2024), and GPT-Neo (1.3B) Black et al. (2021). Rather than using top-p Holtzman et al. (2019) or top-k sampling Fan et al. (2018), which changes the output domain causing potential errors in the influence calculation, we instead employ temperature sampling Ackley et al. (1985) to improve generation quality. During decoding, we chose the temperature parameter $\tau = 0.8$. Each response has length at most $T = 50$ and the number of responses generated for each dataset is $N = 1000$. In section 4.2, three different CID are evaluated: $\lambda = 0.5, 1.0, 1.5$ where $\lambda = 1.5$ is CAD and $\lambda = 1.0$ is just regular decoding (RD).

## 4.2 MAIN RESULTS

Table 1 reports the results on PubMedQA and CNN-DM. The average context influence results $\mathbb{E}[f_{\text{Mem}}(\overline{p}_\theta)]$ for $\lambda = 1.5$ indicate that removing half of the prior knowledge when generating the next token doubles the average context influence for most models over regular decoding. Furthermore, amplifying the context, $\lambda = 1.5$, mostly mitigates hallucination for all models on both datasets. However, there are instances where raising the context influence level can hurt similarity scores,

| CNN-DM | |
|---|---|
| Article | ... Luckily, Japanese can sleep soundly in their beds tonight as the government's top military official earnestly revealed that the country's Air Self Defense Force (ASDF) had never encountered an extraterrestrial unidentified flying object . Responding to a query from flamboyant former wrestler-turned-lawmaker Antonio Inoki, Defense Minister Gen Nakatani told the Diet, Japan's parliament, that his jets had, to date, never come across any UFOs from outer space. ... |
| CAD $\lambda = 1.5$ | Japanese can sleep soundly in their beds tonight as the government's top military official earnestly revealed that the country's Air Self Defense Force (ASDF) had never encountered an extraterrestrial unidentified flying object . |
| RD $\lambda = 1$ | in a interview with Japanese defense minister, politician Antonio Inoki asked the defense minister about aliens and UFOs and the defense minister answered that the Air Self Defense Force ( ASDF ) has never encountered one. |
| $\lambda = 0.5$ | The article discusses the topic of the possible appearance of aliens and their flying vehicles in the skies over Japan. The author of the article recalls that recently there was a flight of a mysterious object in the sky over Japan, which was filmed by the camera of |

Table 2: Qualitative examples from LLaMA 3 using different influence levels of CID. CAD regurgitated the context verbatim and so did RD, but not entirely. For example, both CAD and RD copied UFOs, highlighted in red. However, the response from $\lambda = 0.5$ contains "flying vehicle", highlighted in yellow, which is broadly related to UFO but not exactly contained in the context.

for example, CAD for LLaMA 3 on PubMedQA does not improve over RD, but the faithfulness scores do increase for all models and datasets. Hence, certain models' prior knowledge suffices for answering queries and does not need more context influence. On the other hand, we observed a 10% increase in F1 ROUGE-L and a nearly 1% increase in BERTScore using CAD over RD for LLaMA 3 on CNN-DM, but this caused LLaMA to be influenced by the context 1.5x more. Such an alarming spike demonstrates serious consideration for context influence when mitigating hallucination.

When we reduce the context influence level by selecting $\lambda = 0.5$ so that the decoding is equally split between the prior and posterior distribution, then the average context influence is reduced by more than half for all models. Specifically on PubMedQA, LLaMA 3 is influenced by the context 3x less compared to RD, with ROUGE-L and BERTScore slightly decreased by 8% and 2%, respectively. These decreases in hallucination metrics are worst for abstractive summarization (CNN-DM)— e.g., 68% and 6%— due to the larger reliance on the context. Thus, the influence-hallucination tradeoff isn't as sharp when an LLM can sufficiently rely on its prior knowledge.

Moreover, we observe that LLaMA 3 IT is substantially influenced by the context more than just pre-trained LLaMA 3. This makes sense as LLaMA 3 IT received further training in the form of supervised fine-tuning (SFT) and reinforcement learning with human feedback (RLHF) to align better with prompt answering. These additional steps, SFT and RLHF, help the model utilize the context more when answering queries and, as a result, increase the context influence.

Interestingly, both OPT-1.3B and GPT-Neo 1.3B contain the same number of parameters and roughly follow the model architecture of GPT-3 Brown et al. (2020). Yet, our results show that OPT-1.3B is influenced by the context more than GPT-Neo 1.3B for PubMedQA. Both models are pre-trained on the Pile dataset Gao et al. (2020), which contains abstracts from PubMed, meaning PubMedQA and the Pile intersect. However, OPT only used a subset of the Pile, which does not contain PubMed Abstracts Zhang et al. (2022). So, OPT relies on PubMedQA contexts more than GPT-Neo, which is not as influenced by the context and hence can depend on its prior knowledge more; in other words, OPT's PMI is larger than GPT-Neo's. Therefore, smaller influence by context does not imply smaller privacy risks, as one must consider the public data used for pre-training Tramèr et al. (2022), which could intersect with the provided context.

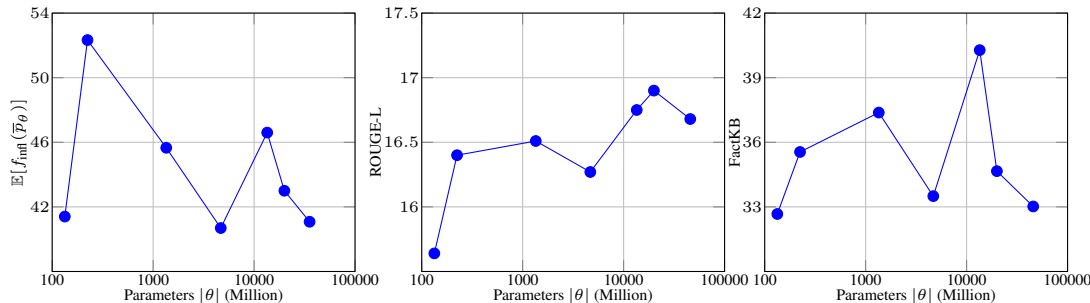

Figure 2: Measuring context influence, ROUGE-L, and FactKB with respect to different OPT parameter sizes $|\theta|$ on PubMedQA where $\lambda = 1.0$.

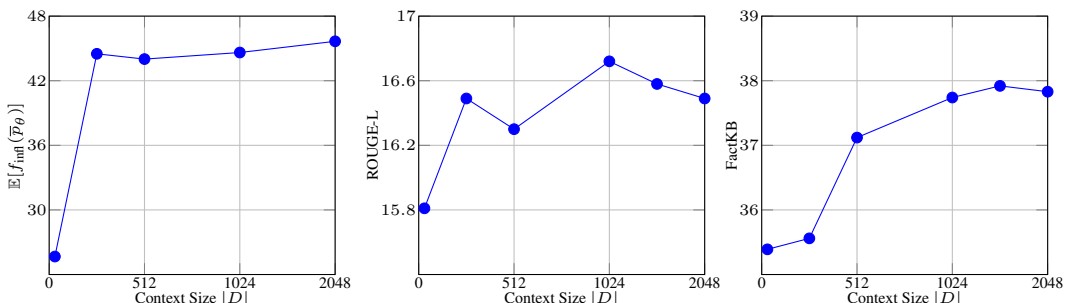

Figure 3: Measuring context influence, ROUGE-L, and FactKB with respect to different PubMedQA context sizes $|D|$ for OPT-1.3B with $\lambda = 1.0$

### 4.3 FURTHER EXPERIMENTAL ANALYSIS

In this section, we explore various parameters, such as model size, context size, and generation length, that could influence a model's propensity to rely on and hallucinate contextual information. Hyperparameters temperature $\tau$ and influence level $\lambda$ are in Appendix D.

**Qualitative examples.** We first qualitatively analyze generations from LLaMA-3 (8B) for CNN-DM in Table 2. We observed that many of CAD's generations are regurgitating the context, highlighting that amplifying the PMI increases surfacing of the provided context. Regular decoding (RD) is also prone to regurgitating context, but not as severely as CAD. In particular, both CAD and RD contain "UFO" in their generations, information derived verbatim from the context rather than prior knowledge. On the other hand, $\lambda = 0.5$ does not contain UFO and instead contains "flying vehicle," a more general entity that is broadly relevant to the context but does not appear verbatim in the context, hence, relying on the prior knowledge. Moreover, CAD and RD can fully capture larger contextual information, such as Japan never encountering UFOs, while $\lambda = 0.5$ can only capture it partially, instead generating "a possible appearance of aliens."

**Model size effect.** Next, we analyze the effect of model size $|\theta|$ on average influence $\mathbb{E}[f_{\text{infl}}(\overline{p}_\theta)]$, ROGUE-L score, and FactKB for CID with regular decoding ($\lambda = 1.0$). We used various sizes– 125M, 350M, 1.3B, 6.7B, 13B, 30B, and 66B– of OPT evaluated on PubMedQA. The results shown in Figure 2 depict a bit of a noisy trend, but generally, larger models are less influenced by the context. This is due to the fact that larger models have a larger capacity to memorize their pre-training data, so they can rely on their prior knowledge more than smaller models. However, very small and medium-sized models, e.g., 125M and 6.7B parameters, seemingly struggle with attending to the context and hence are influenced less by context and hallucinate more (smaller ROGUE-L and FactKB). Consequently, models that rely more on their prior knowledge more are less faithful to the context, as the behavior between the measured average context influence and FactKB is very similar.

**Context size effect.** Additionally, we measured the effect of the context size $|D|$ on average context influence on responses $\mathbb{E}[f_{\text{Mem}}(p_\theta)]$, ROGUE-L score, and FactKB for CID using OPT-1.3B. In this setup, we restrict the model to only the first $|D|$ tokens of context for generation and calculating con-

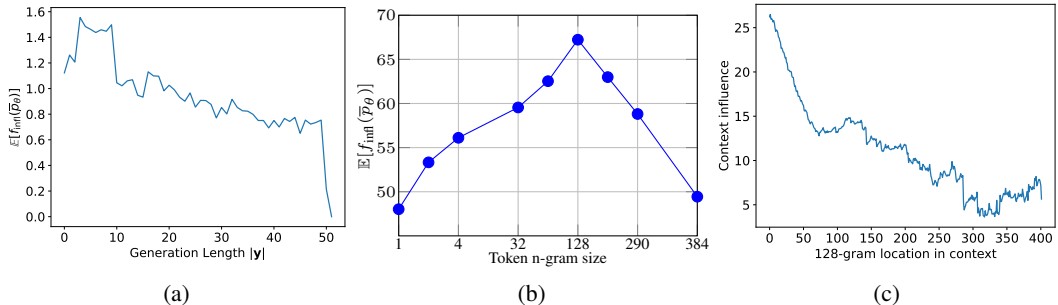

(a)               (b)             (c)

Figure 4: **(a)** Average influence for every token generated in the response. **(b)** Average of token n-grams with the largest influence. **(c)** Average influence of each 128-gram in the context for a response. For all experiments, we used OPT-1.3B and PubMedQA.

text influence and hallucination. Shown in Figure 3, we observe that when the context is extremely small ($\leq 32$) then the LLM is substantially less influenced by the context. The context may not contain enough relevant information to help the model, and hence, it must rely on its prior knowledge, as demonstrated by the lower FactKB. However, as we increase the context size from 32 to 256, the model becomes more influenced by the context and improves response quality via moderate increases in ROUGE-L scores. After $|D| \geq 256$, the model maintains a relatively constant level of context influence, but the model's generations are more faithful to the context (larger FactKB).

**Response Length influenced by context.** Lastly, we measured how far along the prior generation (the size of $\mathbf{y}_{<t}$) affects how much OPT-1.3B is influenced by the context when generating the next token. More precisely, we measure the average context influence of the next token at the $t$-th position $f_{\text{Mem}}(p_\theta, D, D', \mathbf{x}, \mathbf{y}_{<t}, y_t)$ over all generations. As shown in Figure 4a, we observe that the first 10 generated tokens by the model are influenced by the context the most. This is intuitive as the initial response generated by the model is small and nascent; hence, it must rely on the context more for the next token generations. But as the generated response size increases $|\mathbf{y}_t|$, the model relies less on the context and more on its prior knowledge $\theta$ and the current generated response $\mathbf{y}_{<t}$ for generating the next token $y_t$.

### 4.4 TOKEN $n$-GRAM INFLUENCE OF CONTEXT

In this section, we investigate which contiguous subsets of the context during generation had the largest influence, i.e., we compare the output probability with and without a token $n$-gram from the context to measure the influence. Hence, the model still uses the entire context for generations, but token $n$-grams are removed from the context when calculating context influence. This involves iterating through and removing all possible token $n$-grams $D_i$ and recalculating the output distribution without $D_i$ to find the one with the largest influence; more precisely, $f_{\text{infl}}(\overline{p}_\theta, D, D', \mathbf{x}, \mathbf{y}_{<t}, y_t) = \max_{i \in [m]} f_{\text{infl}}(\overline{p}_\theta, D, D_i, \mathbf{x}, \mathbf{y}_{<t}, y_t)$. Due to the possibly large number of possible $n$-grams, we evaluated 100 contexts.

Figure 4b shows the results for various token n-gram influence on PubMedQA for OPT-1.3B with $\lambda = 1.0$. We observe a normal distribution behavior centered at $n = 128$ with the lowest influence values at $n = 1$ and $n = 2048$, suggesting that LLMs are largely influenced by specific sequences of tokens within the contexts. Sequences that are too large contain too much non-relevant information, which is a well-known phenomenon that LLMs struggle with in long contexts Liu et al. (2024). At the same time, those too small do not include enough relevant information. Both cases reduce the influence of the context. Furthermore, we track the average influence of each token 128-gram in the context for a response shown in Figure 4c. We observe that earlier token 128-grams in the context influence the model the most while later ones in the context have less influence. The results suggest that the model is influenced by information located earlier in the context than those located later, which could also be a by-product of relevant information being located earlier in the context.

We qualitatively examined the influence of each uni- and bi-gram within a context using an example generation from Table 2, which is LLaMA 3 with $\lambda = 1.0$. We only selected an excerpt from the original context due to spatial constraints. We created a heatmap-like figure that colors the tokens

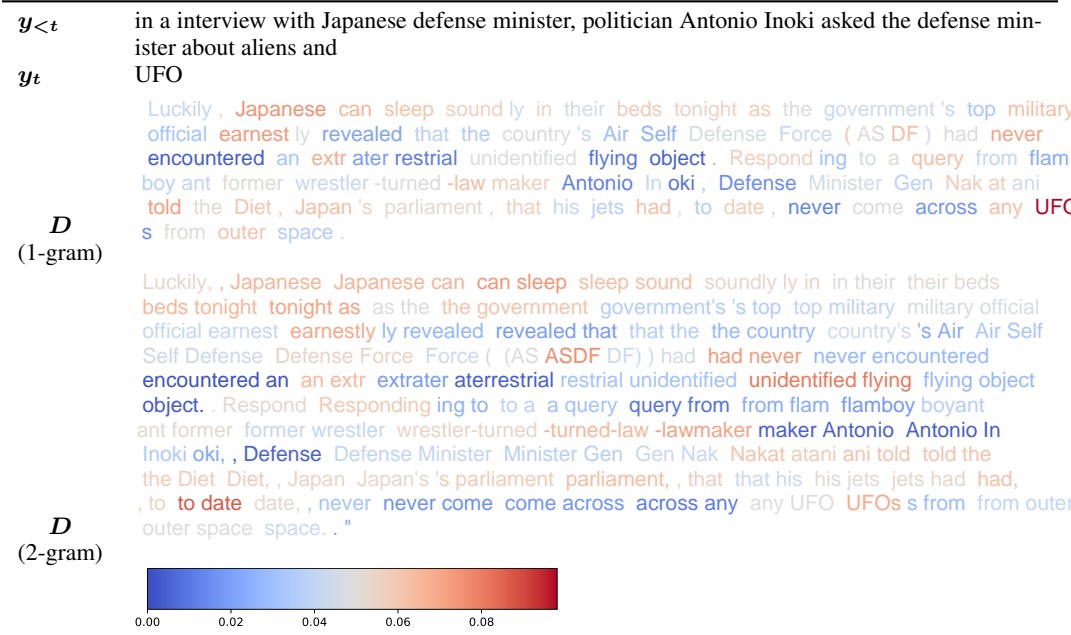

| $y_{<t}$ | in a interview with Japanese defense minister, politician Antonio Inoki asked the defense minister about aliens and |
|---|---|
| $y_t$ | UFO |

Table 3: Using the same qualitative example from LLaMA 3 with $\lambda = 1.0$ in Table 2, we compare the influence by 1-gram and 2-gram tokens of the context $D$ when generating the next token $y_t$ given the previous generations $\mathbf{y}_{<t}$.

of the excerpt based on the uni- and bi-gram influence in Table 3. Additionally, we included the previous generations $\mathbf{y}_{<t}$ and the generated next token $y_t$. In particular, the next token generated is "UFO," and expectedly, the uni-gram from the excerpt with the highest influence is "UFO." Interestingly, we see that words similar to "Japan" also strongly influenced the model, while "flying" and "object" did not. However, for bi-grams, we see that the model was influenced by "unidentified flying" more than "UFOs" and surprisingly is influenced by "to date" the most.

## 5 RELATED WORKS

**Hallucination.** Our work follows prior work on summarization factuality where the response from an LLM conflicts with provided context Maynez et al. (2020); Pagnoni et al. (2021). Several works have explored fine-tuning based methods to improve generation quality Zhu et al. (2020); Cao et al. (2018). Our work focuses on hallucination mitigation during inference Lee et al. (2022). In particular, we focus on techniques that utilize PMI to amplify focus on context rather than prior knowledge Van der Poel et al. (2022); Shi et al. (2023). Our decoding formulation follows from contrastive decoding methods Li et al. (2022); Chuang et al. (2023); Shi et al. (2023) which contrasts an expert's output distribution with an amateur's output distribution. Our work differs from these works in that it connects hallucination with context influence.

**Memorization and Influence.** It has been demonstrated that inadvertent memorization of training data can lead to privacy leakage Carlini et al. (2019); Song & Shmatikov (2019) in the form of extraction attacks Carlini et al. (2021); Thomas et al. (2020). Moreover, the inferential capabilities of LLMs can be exploited to infer undisclosed private information Staab et al. (2023). Hence, there has been a growing body of work analyzing an LLM's memorization capabilities. Label memorization Feldman & Zhang (2020) and its variants, exact memorization Tirumala et al. (2022) and counterfactual memorization Zhang et al. (2023); Lesci et al. (2024), compare how the model performs when trained with and without a particular example from the training set. In contrast, others present more actionable definitions that describe a precise type of memorization Carlini et al. (2021; 2022); Biderman et al. (2024). Works focusing on privacy of contextual information have investigated regurgitation of prompt data by LLMs Priyanshu et al. (2023); Wang et al. (2023) and limited context influence via differential privacy Wu et al. (2023); Tang et al. (2023); Duan et al. (2024). Context

attribution Fernandes et al. (2021); Sarti et al. (2023); Cohen-Wang et al. (2024) is a body of work that looks to attribute a prediction made by an LLM to which parts of the context and share a similar goal to our work.

## 6 DISCUSSION AND CONCLUSION

The goal of our work is primarily motivated by privacy, and, thus, the application of our results hope to inform practitioners of the privacy risks and design solutions with these results in mind. For example, we showed that information appearing earlier in the context influences an LLM more than later ones. Hence, practitioners who want to control the influence of certain sequences can place undesirable/privacy-sensitive ones towards the end of the context/prompt. Additionally, practitioners seeking to privatize LLM generations with respect to a provided context can use Figure 4a to adopt an adaptive privacy level, where the privacy level is strict during the beginning of generating tokens, due to higher influence by the context, then is relaxed as more tokens are generated, since the model can rely on the previous privatized tokens. Lastly, we showed that two LLMs with identical model capacity and architecture can have substantially different influences by the same context, highlighting a greater concern that one must consider not only the prompt but also the pre-training data when measuring the influence of private (contextual) data. It is paramount to identify which parts of the context greatly affect the generations of an LLM to mitigate hallucinations and improve performance, but due to the input regurgitation of LLMs, one must delicately balance context and prior knowledge when generating responses because of the serious utility and privacy implications.

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

## A  PROOF OF THEOREM 3.1

We restate the theorem below:

**Theorem A.1.** Let $\lambda \geq 0$ and $D' = D$. Then the context influence of $D$ with the response $y_t$ generated from CID $\overline{p}_\theta$ (Eq. 4) is

$$f_{\text{infl}}(\overline{p}_\theta, D, D', \mathbf{x}, \mathbf{y}_{<t}, y_t) \leq |\lambda \text{pmi}(p_\theta(y_t; D, \mathbf{x}, \mathbf{y}_{<t}))| . \tag{6}$$

*Proof.* Note that when CID is not provided with the context document, it resorts to sampling purely from the prior distribution regardless of $\lambda$, i.e. $\overline{p}_\theta(y_t|D \setminus D', \mathbf{x}, \mathbf{y}_{<t}) = p_\theta(y_t|\mathbf{x}, \mathbf{y}_{<t})$. Also, let $A = \sum_j \exp[\text{logit}_\theta(y_t = j|D, \mathbf{x}, \mathbf{y}_{<t})]$, $B = \sum_j \exp[\lambda \text{logit}_\theta(y_t = j|D, \mathbf{x}, \mathbf{y}_{<t}) + (1 - \lambda)\text{logit}_\theta(y_t = j|\mathbf{x}, \mathbf{y}_{<t})]$, and $C = \sum_j \exp[\text{logit}_\theta(y_t = j|\mathbf{x}, \mathbf{y}_{<t})]$. Moreover $\log p_\theta(y_t|\cdot) = \text{logit}_\theta(y_t|\cdot) - \log(\sum_j \exp[\text{logit}_\theta(y_t = j|\cdot)])$. Hence

$$\begin{aligned}
f_{\text{infl}}(\overline{p}_\theta, D, D', \mathbf{x}, \mathbf{y}_{<t}, y_t) &= |\log(\overline{p}_\theta(y_t|D, \mathbf{x}_t, \mathbf{y}_{<t}) - \log(\overline{p}_\theta(y_t|D \setminus D', \mathbf{x}, \mathbf{y}_{<t}))| \\
&= |\log(\overline{p}_\theta(y_t|D, \mathbf{x}_t, \mathbf{y}_{<t}) - \log(p_\theta(y_t|\mathbf{x}, \mathbf{y}_{<t}))| \\
&= |\log(p_\theta(y_t|\mathbf{x}_t, \mathbf{y}_{<t}) - \log(\overline{p}_\theta(y_t|D, \mathbf{x}, \mathbf{y}_{<t}))| \\
&= |[\text{logit}_\theta(y_t|\mathbf{x}_t, \mathbf{y}_{<t}) - \log(C)] \\
&\quad - [\lambda \text{logit}_\theta(y_t|D, \mathbf{x}, \mathbf{y}_{<t}) + (1 - \lambda)\text{logit}_\theta(y_t|\mathbf{x}, \mathbf{y}_{<t}) - \log(B)]| \\
&= |\log(B) - \lambda \text{logit}_\theta(y_t|D, \mathbf{x}, \mathbf{y}_{<t}) + \lambda \text{logit}_\theta(y_t|\mathbf{x}, \mathbf{y}_{<t}) - \log(C))| \\
&\leq |\lambda \log(A) + (1 - \lambda)\log(C) - \lambda \text{logit}_\theta(y_t|D, \mathbf{x}, \mathbf{y}_{<t}) \\
&\quad + \lambda \text{logit}_\theta(y_t|\mathbf{x}, \mathbf{y}_{<t}) - \log(C))| \\
&\leq |\lambda \log(A) - \lambda \text{logit}_\theta(y_t|D, \mathbf{x}, \mathbf{y}_{<t}) + \lambda \text{logit}_\theta(y_t|\mathbf{x}, \mathbf{y}_{<t}) - \lambda \log(C))| \\
&= |-\lambda \log(p_\theta(y_t|D, \mathbf{x}, \mathbf{y}_{<t})) + \lambda \log(p_\theta(y_t|\mathbf{x}, \mathbf{y}_{<t}))| \\
&= |\lambda \text{pmi}(y_t; D, \mathbf{x}, \mathbf{y}_{<t})|.
\end{aligned} \tag{7}$$

Eq. 7 is due to the convexity of the logarithm sum of exponentials Boyd & Vandenberghe (2004). $\qquad\square$

## B  PROOF OF THEOREM 3.2

We will now show how CID can satisfy Definition 2.1. First, we are going to slightly modify CID by first selecting $\lambda$ so that we bound the amount of information leaked from a context $D$ when releasing the next token $y_t$. The algorithm can be found in Algorithm 1, which follows from Husain et al. (2020); Flemings et al. (2024).

---

**Algorithm 1** Bounded CID

---

1: **function** $\mathcal{P}(p_\theta, D, \mathbf{x}, \mathbf{y}_{<t}, y_t, \epsilon)$
2: $\quad \lambda_D \leftarrow \frac{\epsilon}{2\text{pmi}(p_\theta(y_t; D, \mathbf{x}, \mathbf{y}_{<t}))}$
3: $\quad \overline{p}_\theta(y_t|D, \mathbf{x}, \mathbf{y}_{<t}) = \text{softmax}[\lambda \text{logit}_\theta(y_t|D, \mathbf{x}, \mathbf{y}_{<t}) + (1 - \lambda)\text{logit}_\theta(y_t|\mathbf{x}, \mathbf{y}_{<t})]$
4: $\quad$ **return** $\overline{p}_\theta(y_t|D, \mathbf{x}, \mathbf{y}_{<t})$
5: **end function**

---

**Theorem B.1.** Let $y_t \sim \mathcal{P}(p_\theta, D, \mathbf{x}, \mathbf{y}_{<t}, y_t, \epsilon)$ be a token generated by the bounded CID from Algorithm 1. Then $y_t$ is $\epsilon$-DP with respect to $D$.

*Proof.* Let $D$ be a dataset and $D' \subseteq D$. Then for any $y_t \in \mathcal{V}$ where $\mathcal{V}$ is the vocabulary of the LLM $p_\theta$, using Definition 2.1, we get the following:

$$\left| \log \left( \frac{y_t \sim \mathcal{P}(p_\theta, D, \mathbf{x}, \mathbf{y}_{<t}, y_t, \epsilon)}{y_t \sim \mathcal{P}(p_\theta, D \setminus D', \mathbf{x}, \mathbf{y}_{<t}, y_t, \epsilon)} \right) \right|$$

$$= \left| \log \left( \frac{\overline{p}_\theta(y_t | D, \mathbf{x}, \mathbf{y}_{<t}, y_t)}{\overline{p}_\theta(y_t | D \setminus D', \mathbf{x}, \mathbf{y}_{<t}, y_t)} \right) \right|$$

$$= \left| \log \left( \frac{\overline{p}_\theta(y_t | D, \mathbf{x}, \mathbf{y}_{<t}, y_t) p_\theta(y_t | \mathbf{x}, \mathbf{y}_{<t})}{\overline{p}_\theta(y_t | D \setminus D', \mathbf{x}, \mathbf{y}_{<t}, y_t) p_\theta(y_t | \mathbf{x}, \mathbf{y}_{<t})} \right) \right|$$

$$= \left| \log \left( \frac{\overline{p}_\theta(y_t | D, \mathbf{x}, \mathbf{y}_{<t}, y_t)}{p_\theta(y_t | \mathbf{x}, \mathbf{y}_{<t})} \right) + \log \left( \frac{p_\theta(y_t | \mathbf{x}, \mathbf{y}_{<t})}{\overline{p}_\theta(y_t | D \setminus D', \mathbf{x}, \mathbf{y}_{<t}, y_t)} \right) \right|$$

$$\leq \left| \log \left( \frac{\overline{p}_\theta(y_t | D, \mathbf{x}, \mathbf{y}_{<t}, y_t)}{p_\theta(y_t | \mathbf{x}, \mathbf{y}_{<t})} \right) \right| + \left| \log \left( \frac{p_\theta(y_t | \mathbf{x}, \mathbf{y}_{<t})}{\overline{p}_\theta(y_t | D \setminus D', \mathbf{x}, \mathbf{y}_{<t}, y_t)} \right) \right| \qquad (8)$$

$$\leq f_{\text{infl}}(\overline{p}_\theta, D, D, \mathbf{x}, \mathbf{y}_{<t}, y_t) + f_{\text{infl}}(\overline{p}_\theta, D', D', \mathbf{x}, \mathbf{y}_{<t}, y_t) \qquad (9)$$

$$\leq |\lambda_D \text{pmi}(p_\theta(y_t; D, \mathbf{x}, \mathbf{y}_{<t}))| + |\lambda_{D'} \text{pmi}(p_\theta(y_t; D', \mathbf{x}, \mathbf{y}_{<t}))| \qquad (10)$$

$$\leq \frac{\epsilon}{2} + \frac{\epsilon}{2} = \epsilon. \qquad (11)$$

Eq. 8 is due to the triangle inequality, Eq. 9 uses our definition of context influence (Eq. 3), Eq. 10 uses Theorem 3.1, and Eq. 11 is from line 2 from Algorithm 1. $\square$

## C  ADDITIONAL EXPERIMENTAL SETUP

| **PubMedQA** | **CNN** |
|---|---|
| Document: Programmed cell death (PCD) is the regulated death of cells within an organism. The lace plant (Aponogeton madagascariensis) produces perforations in its leaves through ... 

 Do mitochondria play a role in remodelling lace plant leaves during programmed cell death? | News article: (CNN)The Palestinian Authority officially became the 123rd member of the International Criminal Court on Wednesday, a step that gives the court jurisdiction over alleged crimes ... 

 Summary of the above news article: |

Figure 5: Example prompts with context used for PubMedQA and CNN where red text is the context $D$ and blue text is the query $x$.

| **PubMedQA** | **CNN** |
|---|---|
| Document: . 

 Do mitochondria play a role in remodelling lace plant leaves during programmed cell death? | News article: . 

 Summary of the above news article: |

Figure 6: Example prompts without context used for PubMedQA and CNN where red text is the context $D$ and blue text is the query $x$.

Figures 5 and 6 illustrate exemplar prompts with and without context used for each dataset in our experiments.

## D  ADDITIONAL EXPERIMENTAL RESULTS

Figure 7 shows the average context influence, ROUGE-L, and FactKB across different temperature values. We observe that as $\tau$ approaches zero, the model is influenced by the context exponentially,

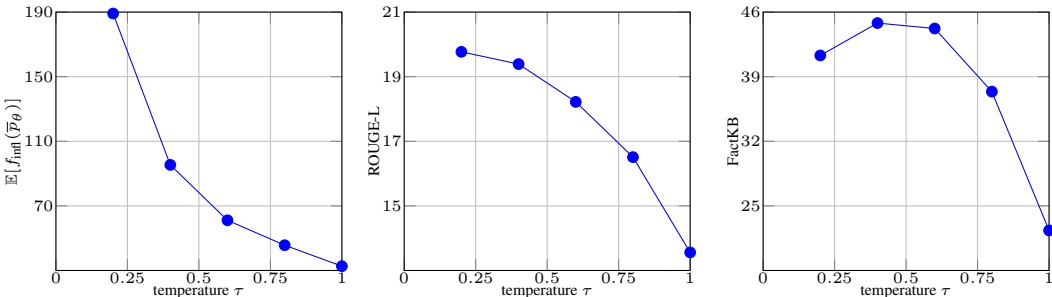

Figure 7: Measuring context influence, ROUGE-L, and FactKB with respect to different temperature $\tau$ values on PubMedQA for OPT-6.7B on PubMedQA using $\lambda = 1.0$

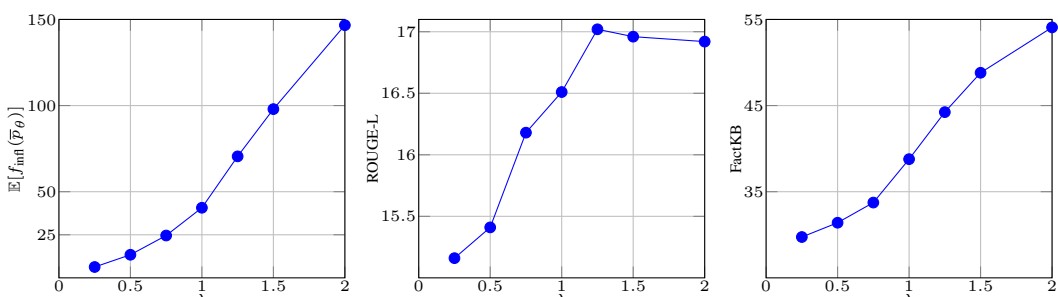

Figure 8: Measuring context influence, ROUGE-L, and FactKB with respect to different $\lambda$ values on PubMedQA for OPT-1.3B.

with moderate improvements in similarity. This is because as $\tau$ approaches zero, the decoding becomes equivalent to argmax, where the token with the highest probability is selected. Hence, there is less entropy in the decoding since the output distributions are sharper, so there is more divergence between the posterior and prior distributions (larger PMI). However, the faithfulness actually decreases once $\tau < 0.4$, demonstrating that less randomness during decoding can result in generations that are not as faithful to the context.

Figure 8 shows the average context influence, ROUGE-L, and FactKB across different context influence levels $\lambda$. Our results suggest that a higher average influence of the context leads to more faithfulness to the context (higher FactKB), but for $\lambda > 1.25$, the similarity of the generated response to the gold response slightly degrades.

