# OpenReview forum: "Characterizing Context Influence and Hallucination in Summarization"
_ICLR.cc/2025/Conference — ICLR 2025 Conference Withdrawn Submission_

### Official Review · Reviewer_Kan7 · 2024-10-22

**Soundness:** 3
**Presentation:** 3
**Contribution:** 3
**Rating:** 6
**Confidence:** 4

**Summary:**

This paper introduces a definition of context influence and presents a decoding method for large language models (LLMs) named context influence decoding (CID). Previous work like [1] amplifies pointwise mutual information (PMI) to mitigate hallucination during decoding. PMI is measured by computing the difference between the probability with and without the context information. This paper argues that PMI could leak privacy since context may contain sensitive information. Specifically, privacy here is defined as the privacy of the prompt (or the context). In response, the authors characterize the context influence, simultaneously considering the privacy of the context.

[1] Shi, W., Han, X., Lewis, M., Tsvetkov, Y., Zettlemoyer, L., & Yih, S. W. T. (2023). Trusting your evidence: Hallucinate less with context-aware decoding. arXiv preprint arXiv:2305.14739.

**Strengths:**

The definition of P-CXMI in Definition 3.1 is an intuitive and useful extension to PMI (or previous P-CXMI), by subtracting a subset of the context and computing their difference.

In Section 3.3, the authors bridge DP and context influence, which I think is novel.

The experiments are adequate and show useful insights for decoding with context. For summarization, the privacy of the context and summarization quality are an important tradeoff.

**Weaknesses:**

Major suggestions.

First, I am worried about the technical contribution of CID. In Equation 5, the authors claim that CID is indeed a linear combination of the prior and posterior logits, which is controlled by an additional temperature parameter. Compared to previous work [1], which proposes to reweight the logit by amplifying posterior $(1+\alpha)logit(y_t|c,x,y_{<t}) - \alpha logit(y_t|x,y_{<t})$, the only difference between this paper and [1] is to introduce a more general parameter $\lambda$, and a temperature parameter. In this way, the authors should consider CID as a simple extension of CAD. Also, from the experimental results, the faithfulness score is not comparable when $\lambda=0.5$, which somehow shows that averaging prior and posterior logits is not a common practice for decoding with context although a smaller $\lambda=0.5$ can lead to a better privacy score.

Second, although the authors define context influence by subtracting a subset of the context instead of subtracting the whole context, from Appendix C, the prompts used in this paper only consider removing the whole context. There seems an additional simplification in the experiment. I understand that removing a subset of a new article or a PubMedQA document is hard to define. However, even a simple experiment that considers this property can help the understanding of context influence.

Minor suggestions.

In Section 3.3, the authors discuss the relationship between CID and differential privacy (DP). I get the concept (e.g., $\epsilon$-DP and the relationship between context influence and DP) only after reading the appendix. I would recommend the authors move some of the definitions and discussion in Appendix B to the main article for a clear presentation. Also, if the authors could accept my first major suggestion, I would recommend that the authors discuss more about privacy in the method section, while keeping CID as an extension of the preliminary.

A typo after Equation 1: (P-XCMI) should be (P-CXMI).

[1] Shi, W., Han, X., Lewis, M., Tsvetkov, Y., Zettlemoyer, L., & Yih, S. W. T. (2023). Trusting your evidence: Hallucinate less with context-aware decoding. arXiv preprint arXiv:2305.14739.

**Questions:**

1. What are the major technical contributions of CID compared to previous work?

2. The authors seem to only evaluate removing all context in the experiment. I would be interested in seeing how removing a subset of context affects the output.

3. Relating context influence with DP is novel, however, is there a way to distinguish the information? In other words, how do the authors determine which part of the information is private and which part benefits the generation quality?

I will raise my score if the authors could address my concerns. I extend my best wishes to the authors for this study.

---

> ### Author Response · Authors · 2024-11-18
> **Response by Authors**
>
> Thank you for your constructive comments. We are glad that you found the idea of the paper interesting and impactful. We address your concerns below:
>
> >In this way, the authors should consider CID as a simple extension of CAD.
>
> We agree that CID is a simple extension of CAD. We have revised Section 3.2 and the contributions paragraph in Section 1 to reflect this better.
>
> > Also, from the experimental results, the faithfulness score is not comparable when $\lambda=0.5$, which somehow shows that averaging prior and posterior logits is not a common practice for decoding with context although a smaller $\lambda=0.5$ can lead to a better privacy score.
>
> Yes, our experimental results demonstrate a tradeoff between faithfulnes and privacy, where a better privacy score leads to a worse faithfulnes score. This tradeoff makes sense because the generations are relying less on the context ($\lambda=0.5$) and hence will be less faithful to the context. But the privacy will improve since less privacy of the context is likely to be leaked from the generations.
>
> > the prompts used in this paper only consider removing the whole context.
>
> We actually did perform experiments that consider removing subsets of the context, which can be found in section 4.4. We accomplished this by removing n-grams of the context (contiguous subsets) and measuring the n-gram influence. Please look at the results in this section to help understand context influence. However, for the main results (Section 4.2), we only looked at context influence for an entire context. In our revision, we have added that the prompts used in Appendix C are only for the main results.
>
> >I would recommend the authors move some of the definitions and discussion in Appendix B to the main article for a clear presentation. Also, if the authors could accept my first major suggestion, I would recommend that the authors discuss more about privacy in the method section, while keeping CID as an extension of the preliminary.
>
> We appreciate the reviewer's suggestions and accept them into our revision. In particular, we moved the DP definition to the preliminaries, added more discussion about DP in section 3.3, and removed some discussion of CID to keep it more as an extension of CAD.
>
> >What are the major technical contributions of CID compared to previous work?
>
> Prior work has only focused on analyzing privacy leakage of pre-training data via memorization. Other works have looked into context attribution by analyzing contextual information, but is a different goal from analyzing privacy leakage. Some works have looked into privacy leakage of RAG, but they did not use context influence or DP and instead employed membership inference attacks. We are the first to define a measure for privacy leakage of contextual information via context influence and connect it with DP. We also introduced and proved a context influence-hallucination tradeoff. And finally, we are also the first work to look at how other factors such as model parameters, context size, response length, and n-grams of context affect privacy of context.
>
> >The authors seem to only evaluate removing all context in the experiment. I would be interested in seeing how removing a subset of context affects the output.
>
> Please see section 4.4 which contains results on how removing a subset of the context affects the output.
>
> >Relating context influence with DP is novel, however, is there a way to distinguish the information? In other words, how do the authors determine which part of the information is private and which part benefits the generation quality?
>
> This is a really good question that needs much more future work: Determining which subset of a context is private is a more realistic setup, as we could treat some parts of the context as private and the rest to be public information. However, we surmise at this time that this level of separation is difficult to achieve in an automated way without some sort of human intervention. And the question of which part benefits the generation quality is another line of work called context attribution, which is a different goal of ours but also relevant. Our n-gram influence results does give some insight into which parts benefit the generation quality. In particular, 128-gram subsets of PubMedQA documents have the highest influence on the model compared to other n-gram sizes.

---

> > ### Comment · Reviewer_Kan7 · 2024-11-20
> >
> > I would like to appreciate the response by the author team. They generally solve all my concerns. Below are my further ideas. Please kindly take my ideas as suggestions not critical comments.
> >
> > ```
> > Section 4.4
> > ```
> >
> > I have read section 4.4 again, and now I understand that the authors want to discuss the context influence by removing n-gram subsets. Also, the parameters of the LLM and the parameters of the n-gram subsets are explored. In this way, the results between sections (4.2-4.3) and section 4.4 are more likely to be parallel, not in a main-sub setting. My initial confusion mainly comes from the prompt you provided in Appendix (in the first manuscript) and these titles.
> >
> > ```
> > Distinguish between private and useful contexts.
> > ```
> >
> > My last question seems a little off-topic to this study although it is a practical question. I thank the authors for the reply and do not think it is possible to involve human intervention in all research projects.
> >
> > I have raised my score.

---

> > > ### Author Response · Authors · 2024-11-20
> > > **Response by Authors**
> > >
> > > We appreciate the reviewer's suggestions and for raising their score.

---

### Official Review · Reviewer_F83x · 2024-11-02

**Soundness:** 2
**Presentation:** 2
**Contribution:** 2
**Rating:** 5
**Confidence:** 4

**Summary:**

This work studies the relationship between hallucination (generating content that contradicts context) and privacy risks from input regurgitation in LLMs. The authors propose a formal definition of "context influence" and introduce CID, demonstrating that amplifying context to reduce hallucination leads to increased context influence and potential privacy leakage. Through experiments using multiple models and datasets, they show that improving generation quality comes at the cost of significantly increased context influence, while also analyzing how factors like model capacity, context size, and response length affect this tradeoff between hallucination and privacy.

**Strengths:**

The problem of context leakage is very interesting, especially since RAG models are being more used in industry now. Theory statements seem correct.

**Weaknesses:**

## Sections 1 and 2

I'm familiar with hallucination/trustworthiness in LLMs, but not with the specific paper(s) this works draws ideas from. So it was particularly hard for me to follow the jargons used, I wasn't able to precisely infer the contributions from it. In particular, I suggest the authors change the presentation incorporating a figure with a running example of the problem. Some of the sentences are not very accurate, e.g.

> With the knowledge obtained during pre-training Devlin et al. (2018); Radford et al. (2019), due to the exponential scaling of the transformer architecture Vaswani (2017) and pre-training data Chowdhery et al. (2023), LLMs display an In Context Learning (ICL) ability to further improve on various downstream tasks without additional training by supplementing prompts with relevant context

I don't think we understand the emergence of ICL that well ---in specific the citations don't claim that. We have a few promising hypotheses in other papers, but nothing that is as strong as the above claim. The authors can just say it's a well-documented emergent behavior in LLMs.

It's very hard to understand the problem setup and motivation in the first two pages. For instance, the words "context" and "hallucination" are consistently used without an appropriate definition. The paper is heavily based on Shi's contribution, which I was not familiar with. I read it and then understood section 2, but I think the authors need to rethink how they present it, making it more self-contained. An important comment to make about Eq 2 is that it biases the distribution by making the model more confident when the document posterior diverges from the prior. I think this comment can compress most of the text there.

## Section 3
>  we present a slightly more granular definition of P-CXMI

P-CXMI is only mentioned once in the paper, I still don't know what it is.
Def 3.1:
Documents were defined as vectors, not sets, do the authors mean that $D'$ is a substring of $D$?
What's $f_{Mem}$?

The author's main contribution is Eq 5. I'm not sure I see the novelty here other than simply down-weighing the context influence in the decoding.

> Since sampling from a probability distribution inherently induces privacy

This statement is wrong. In general privacy is saying a divergence between the distribution coming from one dataset to the distribution coming from an adjacent dataset is small. These are very different things, simply having a distribution isn't enough for privacy, e.g. dirac delta.

## Section 4

I had a hard time understanding the results. What's the main question the authors are trying to answer? It seems that the results just show how Eq 2, from Shi, is a good strategy to ground the model's faithfulness. What's the takeaway with respect to the authors method? It's not clear to me.  From the abstract I can tell you want to show that grounding the answer costs leaking context, but my comment for here is that this is not clearly highlighted in the section.

**Questions:**

I have a few questions in the weaknesses box, please refer to them. Here's some others:
- I don't think I understand the role of theorem 3.1. Can you explain to me its relevance? From the definition of pmi in Eq 2 it's straightforward to see that it enforces the context influence, since it was designed for it. Thus, as you increase the penalty, the more you'll concentrate.
- What is the takeaway of section 3.3 if you do not know $\lambda^*$? You can always tweak a model to achieve DP with some specified, unknown, parameter.
- Why do you focus on summarization? It seems that the focus is only because the experiments were run on summarization.

---

> ### Author Response · Authors · 2024-11-18
> **Response by Authors Part 1**
>
> Thank you for your constructive comments. We are glad that you found the idea of the paper interesting and impactful. We address your concerns below:
>
> > In particular, I suggest the authors change the presentation incorporating a figure with a running example of the problem.
>
> We have included additional information relating Figure 1 to the main text with a running example in our revision.
>
> > The authors can just say it's a well-documented emergent behavior in LLMs.
>
> We have revised the aforementioned sentence to reflect the reviewer's comment.
>
> > For instance, the words "context" and "hallucination" are consistently used without an appropriate definition
>
> We define these terms in the first and second paragraph of the perliminaries. From our manuscript, we define $\textbf{context}$ "$D=[d_1,...,d_n]$ be some context document/text" and $\textbf{hallucination}$ "It is possible that the resulting response $\mathbf{y}$ contains fictitious information--- i.e., $\mathbf{y}$ is not supported by $D$--- which we deem as a hallucination by the LLM." Please let us know if there is any confusion here.
>
> > but I think the authors need to rethink how they present it, making it more self-contained.
>
> We felt our description of CAD sufficied for understanding the methodology of our work, since it was carefully paraphrased from Shi et al., 2023. We would value your input in what we can cut/add to make the CAD discussion more precisely self-contained.
>
> > An important comment to make about Eq 2 is that it biases the distribution by making the model more confident when the document posterior diverges from the prior. I think this comment can compress most of the text there.
>
> Entropy, which is a measure of uncertainty, can be still large even when applying Eq. 2. Actually, it can be the case that the model's prior output is quite confident, but its document posterior output is not confident. Van der Poel et al., 2022 included entropy in their formulation of Eq. 2 to account for this case. What Eq. 2 does is bias the LLM towards the posterior when it diverges from the prior without necessarily making the model more confident.
>
> > P-CXMI is only mentioned once in the paper, I still don't know what it is.
>
> We presented P-XCMI in Eq. 1 of the preliminaries section. To clarify for the confusion, we have included a reference to Eq. 1 prior to introducing Def 3.1 in our revision.
>
> > do the authors mean that $D'$ is a substring of $D$? What's $f_{\text{Mem}}$?
>
> Yes, $D'$ is a substring of $D$. And $f_{\text{Mem}}$ is a typo. We have revised these accordingly in our revision.
>
> > The author's main contribution is Eq 5. I'm not sure I see the novelty here other than simply down-weighing the context influence in the decoding.
>
> Eq. 6 (and Thm 3.1) are the primary contributions of our work. Eq. 5 only reformulates the CAD framework from Shi et al., 2023, so we can directly appply that formulation in our context influence calculation. We have made this point more clear in Section 3.2 of our updated revision.
>
> > This statement is wrong. In general privacy is saying a divergence between the distribution coming from one dataset to the distribution coming from an adjacent dataset is small
>
> We do not disagree with the broader definition of privacy you provided. But our statement merely follows prior work (Wang et al., 2015) which states that obtaining a sample from a distribution can inherently provide privacy (since sampling is not deterministic and instead stochastic). However, sampling alone does not bound the privacy leakage, and hence, the sample is not differentially private until we make more modifications to the CID framework (as is done in Appendix B).
>
> >What's the main question the authors are trying to answer? ... What's the takeaway with respect to the authors method?
>
> The key takeaway from the main results is that relying too much on the context for mitigating hallucination leads to higher risk of privacy leakage. We theoretically demonstrated this with Theorem 3.1 and we experimentally showed this in section 4.2, where increasing the F1 ROUGE-L by 10\% caused LLaMA to be influenced by the context 1.5x more. Table 2 qualatatively showed that this increase in context influence leads to regurgitation of the context, a direct privacy leakage. Additionally, we can make this connection in the experimental results between context influence and privacy because of section 3.3, which was not addressed in prior works. Moreover, the experimental analysis in Section 4.3 explored how context influence can be affected by important hyperparameters, such as model size, context size, and generation length. We discuss the privacy implications of these results in Section 6.

---

> > ### Author Response · Authors · 2024-11-18
> > **Reponse by Authors Part 2**
> >
> > > I don't think I understand the role of theorem 3.1. Can you explain to me its relevance?
> >
> > Theorem 3.1 is highlighting a context influence-hallucination tradeoff, where the context influence is bounded by the pmi, which is a fixed measure of how much the model relies on the context given the query and current response, and $\lambda$, which controls  how much one wants to mitigate context-conflicting hallucination by factoring out prior knowledge. We have added this discussion to our revision to better describe the role of theorem 3.1.
> >
> > >What is the takeaway of section 3.3 if you do not know $\lambda^*$?
> >
> > The takeaway is that you can change the influence of the context on the output of an LLM, but you may not achieve DP unless you select the right $\lambda^*$ such that the DP definition is satisfied. We have included this discussion in section 3.3 of our revision.
> >
> > > Why do you focus on summarization?
> >
> > Our research is focused on context-confliciting hallucination, and summarization tasks involve generating a response with respect to a provided context. With summarization, there is an interesting tradeoff between an LLM's prior knowledge and the contextual information provided in the prompt.

---

> > > ### Author Response · Authors · 2024-11-25
> > > **A Gentle Reminder**
> > >
> > > Dear Reviewer F83x,
> > >
> > > We have carefully considered your initial feedback in our response. Please let us know if we have properly addressed your concerns, and if so, please kindly consider raising your intial score. We are more than happy to answer any further questions.
> > >
> > > Thank you for your time and effort to review our work!

---

> > > > ### Comment · Reviewer_F83x · 2024-11-25
> > > > **Response**
> > > >
> > > > Thank you for the rebuttal. The authors clarified my concerns a bit, so I raised the score to 5, but I still think the contributions are scattered and the paper could benefit from clearer experiments ---since the theoretical contributions are still marginal to me.

---

> > > > > ### Author Response · Authors · 2024-11-25
> > > > > **Reponse by Authors**
> > > > >
> > > > > We thank the reviewer for their prompt response and for raising their score. We would like to follow up and ask what experiments the reviewer would want to see in the paper and how they want us to better assemble the contributions, given the remaining rebuttal window.

---

### Official Review · Reviewer_cUjU · 2024-11-04

**Soundness:** 3
**Presentation:** 3
**Contribution:** 3
**Rating:** 5
**Confidence:** 4

**Summary:**

This paper studies the relationship between context influence and privacy leakage (e.g., PIIs) in LLMs. Specifically, context influence and context influence decoding (CID) were introduced and shown to give a lower bound of the private leakage of CID.

**Strengths:**

Strengths:
- The paper is very well written and easy to read.
- The topic is quite timely and interesting.
- The method is sound and summarization experiments (e.g., context influence, etc.) have been conducted on two datasets to show the effectiveness of the proposed method.
- Mathematical proofs and qualitative examples are provided to back up the proposed idea and results.

**Weaknesses:**

Weaknesses:
- My overall assessment of the paper is mostly positive, but I do have the following concern/question re the privacy part of the paper:

Q. The main focus of the privacy part of the paper is on the fact that LLM can inadvertently regurgitate private information in the prompt. However there is no evaluation of the method whatsoever nor a relevant privacy metric in an adversarial learning setting (e.g., private attribute inference attacker [1])  that quantifies how well the proposed method is able to protect users from such attacks. In fact, except for the connection between the proposed method and DP mentioned in the appendix, I don’t see any other relevant information in the paper. I’d like to see some experiments and evals similar to [1] on different LLMs’ summary outputs (especially those from smaller ones) on both datasets.

[1] Privacy-Aware Recommendation with Private-Attribute Protection using Adversarial Learning, 2019.

**Questions:**

Please see the weaknesses part.

---

> ### Author Response · Authors · 2024-11-18
> **Response by Authors**
>
> Thank you for your constructive comments. We are glad that you found the idea of the paper interesting and impactful. We address your concerns below:
>
> >there is no evaluation of the method whatsoever nor a relevant privacy metric in an adversarial learning setting (e.g., private attribute inference attacker [1]) that quantifies how well the proposed method is able to protect users from such attacks.
>
>
> Thanks for sharing this line of work. Our work is motivated by Differentially Privacy (DP)-centric privacy leakage gurantees which provide a bound on the amount of information leaked, without any assumptions made by the attacker. In Section 3.3, we showed that our context influence definition is intimately connected with DP. More specifically, conext influence *exactly* follows the definition of DP except that we have fixed the private dataset $D$, the neighboring dataset $D'$, and the output $y_t$. Context influence measures the output change if one were to remove a substring from the context. Large output change means the model relied heavily on the substring, which is a privacy risk. Hence context influence gives a strong lower-bound on the $\epsilon$ value used to measure privacy leakage with DP. Given this, we believe that  Context Influence robustly measures the privacy leakage of a provided context/document when sampling from an LLM, compared to privacy attribute inference [1] which instantiates an attacker by using adversary learning to measure a particular privacy leakage using attribute inference with AUC. Morevoer, it is not entirely clear how we can use the same attacker in [1] to our setting, but it can be an interesting follow up work.
>
> > In fact, except for the connection between the proposed method and DP mentioned in the appendix, I don’t see any other relevant information in the paper
>
> We have added more discussion about context influence's connection with differential privacy in Section 3.1 and 3.3 of our revision, as well moved relevant privacy definitions to the preliminaries, which hopefully makes the privacy part of our paper more clear.

---

> > ### Author Response · Authors · 2024-11-25
> > **A Gentle Reminder**
> >
> > Dear Reviewer cUjU,
> >
> > We have carefully considered your initial feedback in our response. Please let us know if we have properly addressed your concerns, and if so, please kindly consider raising your intial score. We are more than happy to answer any further questions.
> >
> > Thank you for your time and effort to review our work!

---

> > > ### Comment · Area_Chair_ybaH · 2024-11-26
> > >
> > > Dear reviewer cUjU,
> > >
> > > Could you please respond to authors' rebuttal and see if you would like to update your review? Thanks very much!
> > >
> > > AC

---

> > > > ### Comment · Reviewer_cUjU · 2024-12-02
> > > > **Thanks for the reminder**
> > > >
> > > > I would like to thank authors for their response, clarification RE DP and the revisions and also apologize for my late response. Having read the response, and other reviewers' feedback once again, unfortunately I am still not convinced about the eval part and would like to keep my score. Best of luck to the authors!

---

### Note · Authors · 2024-12-03

**Comment:**

We have decided to withdraw the paper to improve it further.

**Withdrawal Confirmation:**

I have read and agree with the venue's withdrawal policy on behalf of myself and my co-authors.